# High-Voltage FDS of Thermally Aged XLPE Cable and Its Correlation with Physicochemical Properties

**DOI:** 10.3390/polym14173519

**Published:** 2022-08-27

**Authors:** Haoyue Wang, Maolun Sun, Kaijie Zhao, Xiaowei Wang, Qilong Xu, Wei Wang, Chengrong Li

**Affiliations:** 1State Key Laboratory of Alternate Electrical Power System with Renewable Energy Sources, North China Electric Power University, Beijing 102206, China; 2State Grid Shandong Electric Power Company, Heze 274000, China; 3Hohhot Power Supply Bureau of Inner Mongolia Power (Group) Co., Ltd., Hohhot 010010, China

**Keywords:** crosslinked polyethylene (XLPE) cable, thermal aging, frequency domain spectroscopy (FDS), physicochemical properties, diagnosis

## Abstract

This paper aims to investigate the influence of thermal aging on a crosslinked polyethylene (XLPE) cable, and the relationships between the macroscopical high-voltage dielectric and the microscopical physicochemical properties are also elucidated. To better simulate thermal aging under working condition, the medium-voltage-level cable is subjected to accelerated inner thermal aging for different aging times. Then, high-voltage frequency domain spectroscopy (FDS) (cable sample) and analyses of microscopic physical and chemical properties (sampling from the cable), including Fourier transform infrared spectroscopy (FTIR), X-ray diffraction (XRD), and elongation at the break (EAB), are conducted at different cable aging stages. The dielectric test results show that after a certain aging time, the high-voltage FDS curves of the cable have layered characteristics, and this phenomenon is more obvious as the aging degree increases. Moreover, the slope and the integral of the high-voltage FDS curves rise with aging time. The mechanism is deduced by the physicochemical results that thermo-oxidative aging results in increasing polar groups and dislocation defects in the crystal region, which leads to the above phenomenon. On the one hand, the appearance of polar groups increases the density of the dipole. On the other hand, the destruction of the crystal region increases the probability and amplitude of dipole reversal. In addition, the breaking of molecular bonds and the increase in the amorphous phase also reduce the rigidity of the XLPE molecular main chain. The above factors lead to obvious delamination and larger dielectric parameters of the thermally aged cable. Finally, according to the experimental results, an on-site diagnosis method of cable insulation thermal aging based on high-voltage FDS is discussed.

## 1. Introduction

Crosslinked polyethylene (XLPE) is widely used as an electrical insulator in extruded power cables [1]. However, it is now understood that XLPE cables experience aging when subjected to various stresses (electrical, thermal, mechanical, etc.) in service. Among them, the thermal constraint is considered as the most severe factors causing irreversible damage to the electrical insulation [2,3]. The damage strongly reduces the performance of the XLPE cable and its physical morphology, potentially leading to failure.

The morphological changes occur in XLPE due to thermal aging. Many researchers have focused on physicochemical analyses of thermal aging caused by parameter changes. The representative of these are melting point [4], crystallinity [5], infrared carbonyl absorbance [6], oxidation induction time, electrical strength, tensile strength [7], and elongation at break (EAB) [8]. However, using the above properties to monitor the aging states and evaluate the conditions of the cable insulation still has quite a few problems. First of all, some methods require sophisticated equipment, which are expensive and time consuming. More importantly, most of these methods need to slice the cable, which is destructive to the cable itself, and therefore limits their application in the field.

Recently, extensive attention has been paid to on-site methodological research for the monitoring and assessment of power cables. The insulation status evaluation of the XLPE power cable should use non-destructive methods. Nowadays, research mainly focuses on partial discharges (PD) and frequency domain spectroscopy (FDS) detection to assess insulation materials [9,10]. Some researchers have found that PD is not sensitive to thermal aging, while the dielectric parameters, i.e., dielectric loss tangent and dielectric permittivity, are closely related with the cable aging type and degree [11,12]. FDS is commonly used to estimate water content in the paper insulation of high-voltage transformers [13,14]. Some researchers have revealed that FDS results are sensitive to the variation of water and aging products, and the aging of the pressboard sample. Therefore, the FDS results can reflect water and aging states, and thus, the possible insulation condition of the transformer. At present, some early-stage dielectric changing rules during the XLPE sample aging process are obtained [15,16]. For example, the area of the FDS curve of the XLPE sample increases during the thermal aging process, and the low-frequency curve has stronger regularity than that of the high-frequency range. In some specific frequency ranges, the loss peaks can appear or disappear with the deepening of the aging degree. However, the research on FDS detection for XLPE cable thermal aging is mainly based on phenomenon description, and there is no effective judgment standard. Moreover, the existing research on FDS detection for thermal aging are mainly focused on the low-voltage test of XLPE materials and miniature cable samples, and due to the low testing voltage, the detection sensitivity and anti-interference are reduced.

Meanwhile, high-voltage FDS shows great potential in cable aging detection [17,18]. Research has found that aging cables have some special dielectric properties under high voltage that do not exist under low voltage [19]. High-voltage FDS has been used to realize the water tree aging detection of power cable. It has been found that the tanδ-f curve of the aging cable rises with the increasing detection voltage, which can be used as the diagnosis basis of water tree aging. This result proves the high sensitivity of high-voltage FDS detection for cable aging. Considering that dielectric detection should not cause secondary damage to the cable, CIGRE stipulates that the maximum detection voltage of the dielectric spectrum detection should not exceed 2*U*_0_ [20]. Therefore, combined with the existing research results, the highest detection voltage of high-voltage FDS should be between *U*_0_–2 *U*_0_.

In this paper, the inner heating method is used to better simulate the overheating aging process in the actual cable operation, and an FDS system with a maximum voltage of *U*_0_ and a frequency range of 0.01 Hz–0.1 Hz is used for the thermally aged 8.7/10 kV cable measurement. The aim in selecting the frequency band is to balance the detection time and accuracy. Previous studies have shown that a decrease in the detection frequency can more sensitively reflect the material aging state [21], but it has longer detection time (for example, the detection frequency of 0.001 Hz requires 1000 s for one detection cycle). Setting the detection frequency band at 0.01 Hz–0.1 Hz can not only improve the detection sensitivity, but also realize the fast detection. We observe, summarize, and analyze the high-voltage dielectric properties variation of power cable at different voltage levels during the thermal aging process. At the same time, this study also aims at elucidating the correlations between structural changes (physical, chemical, and mechanical changes) of XLPE material and the dielectric properties variation of XLPE cable during thermal aging. This leads to a better understanding of changes in dielectric properties and the mechanism of material degradation. Finally, the parameters are extracted from the high-voltage FDS to realize the diagnosis and evaluation of cable thermal aging.

The rest of this article is organized as follows. Section 2 describes the sample preparation process and experimental methods. The high-voltage FDS and microscopical physicochemical test results are displayed in Section 3, and the mechanism for the changes in high-voltage FDS during the thermal aging process are analyzed. The relationships between the macroscopical high-voltage dielectric and the microscopical physicochemical properties are also elucidated. Furthermore, the possibility and feasibility of using high-voltage FDS as a field test method for cable thermal aging is also discussed in this section. Finally, the conclusions are stated in Section 4.

## 2. Experimental Setup

### 2.1. Sample Preparation

Figure 1 shows the XLPE cable sample under pretreatment. The YJV-8.7/10 kV XLPE cable from Yuandong company, Shandong, China (insulation thickness 4.5 mm, 4.2 m) is used for the thermal aging experiment and the subsequent tests. The outer semiconducting layer near the copper joints is peeled off by 5 cm to 10 cm for safety consideration. Copper joints are installed at both ends of the sample for better connection.

The overall thermal aging of cables in operation is generally caused by overcurrent in the conductors. In order to better simulate the overheating condition under an operation situation, the internal heating method is used in the accelerated thermal aging experiment. The overall accelerated thermal aging system (composed of large current generator and temperature monitoring system) is shown in Figure 2. The large current generator is used to heat the cable conductor. In order to control the heating temperature, the temperature acquisition system is used to monitor the temperature of both the cable sample and short temperature measurement sample (60 cm). From Figure 2, ① and ② are temperature-measuring thermocouples. ① accesses to the copper core of the temperature measurement cable sample; ② and ③ measure the temperature of the outer sheath of both the temperature measurement cable and the actual aging sample. According to IEC-60502, the thermal aging treatment temperature of XLPE cable insulation is 135 °C. Thus, the heating temperature is set to be 135 °C [22]. The conductor temperature can be observed at any time through the temperature-measuring thermocouple to determine that the temperature at the most strongly heated part of the cable insulation is 135 °C.

In this experiment, the aged cable insulation is cut into the inner, middle, and outer layer strip samples using a crosscutting machine. The 1.5 mm XLPE layer is cut by the crosscutting machine first, then three XLPE strips 0.6 mm in thickness are cut from the remaining 3 mm layer. Finally, the dumbbell samples are formed using a dumbbell die machine for mechanical, FTIR and XRD test. Figure 3 shows the schematic diagram of the cable crosscutting. The samples required for microscopical and physicochemical measurement (FTIR, XRD, and mechanical test) are taken from the most strongly aging part of the cable (inner slice).

The aging time lasts 1536 h (64 days). The overall thermal aged cable is tested by high-voltage FDS at the aging time points of 48 h, 96 h, 192 h, 384 h, 768 h, and 1536 h, respectively, and the material is also collected for the mechanical properties test, FTIR, and XRD at the same time points.

### 2.2. Measurements Carried Out

#### 2.2.1. Macroscopic High-Voltage FDS Measurement

The high-voltage FDS testing platform is composed of a high-voltage FDS detection system and cable sample, as shown in Figure 4. The cable core is connected to the output end of the detection system, and the cable ground wire is connected to the current detection end. The FDS is measured in the following order: start at 0.25 *U*_0_, then increase by intervals of 0.25 *U*_0_, gradually increase the detection voltage to *U*_0_. The detection frequency sequence is 0.01 Hz–0.02 Hz–0.05 Hz–0.1 Hz at each detection voltage.

#### 2.2.2. Microscopical Physicochemical Measurement

In this experiment, we use the EAB parameter to characterize the XLPE mechanical properties. The dumbbell-shaped specimens are formed following the method described in Section 2.1 and tested using a tensile testing machine. Due to the sensitivity of EAB to the surface condition of specimens, three dumbbell-shaped samples are tested in each aged group.

FTIR analysis evaluates the changes in the microstructure and physicochemical properties of the cable insulation, as well as cable stability. A Perkin-Elmer Cetus instrument (Norwalk, CT) is used. First, 1 to 2 mg powder is ground from the inner layer sample. The sample powder is then fully mixed with KBr and placed into the mold. Finally, the film required for FTIR analysis is produced by the pressing machine. 

XRD is used to determine the crystalline structure changes in the XLPE insulation. The test equipment used here was RIGAKU. The test samples are all 20 mm × 20 mm × 0.6 mm. The scanning range is 10° to 30° and scanning rate is 4°/min.

## 3. Results and Discussion

### 3.1. Microscopical Physicochemical Results

#### 3.1.1. Test Results of Mechanical and Physicochemical Properties

The mean value of the EABs of three specimens and the sample variance versus aging time is shown in Figure 5; all data are from the most severely aged part (inner layer) of the cable insulation. Through the analysis of variance, it is found that there is no significant difference in the data of each group, meaning good consistency. From Figure 5, the test results of mechanical properties show that the overall change trend of XLPE material EAB has a good monotonic relationship with its aging degree. When the aging degree is mild (48–96 h), the change in EAB is relatively gentle. After aging for a period of time, the EAB begins to decrease, reaching a moderate stage of aging (192–384 h). At the severe aging stage (768–1536 h), due to the increase in chemical reaction rate, the decrease rate of EAB accelerates and rapidly decreases to the attention value (EAB 75%), and then the failure value (EAB 50%).

The FTIR spectra of cable insulation versus aging time are shown in Figure 6. At the early aging stage (thermal aging 48–96 h), no significant change in the FTIR spectra of XLPE is observed except for the slight fluctuation in 1020–1380 cm^−1^. This feature could be explained by the fact that the antioxidant has not been consumed completely during this period. Additionally, the fluctuation of 1020–1380 cm^−1^ band should be related to the consumption process of the antioxidants. These infrared absorption peaks appear at about 1078, 1159, and 1300 cm^−1^, respectively [4]. After 192 h of thermal aging, at the middle aging stage, the antioxidant is gradually exhausted; it can be seen that a new absorption peak appears at the wavenumber 1720 cm^−1^, and the intensity of this absorption peak increases with the increase in thermal aging time. This type of stretching vibration at 1720 cm^−1^ corresponds to carbonyl (-C=O), showing that carbonyl is produced in crosslinked polyethylene during thermal aging; the carbonyl absorption peak can be used as the characteristic peak characterizing thermal oxygen aging. At the same time, it can also be seen from the spectrum that the intensity of the absorption band at the wavenumber 2750–3000 cm ^−1^ region and the intensity of the absorption peak at 720 cm^−1^ and 1460 cm^−1^ are significantly reduced, which is related to the increase in the carbonyl peak intensity and the emergence of the ether absorption peak at 1169 cm^−1^ (the vibration frequency is 1000–1300 cm^−1^). A large amount of carbonyl production leads to the reduction in the number of methyl groups (-CH, -CH_2_, and -CH_3_). The carbonyl absorption peak increased drastically at wavenumber 1720 cm^−1^ after thermally aged for 784 h (i.e., at the late aging stage), indicating that the cable has undergone chain breaking and oxidation reaction during this period [8].

The XRD spectra of XLPE insulation at different aging stages are shown in Figure 7. The spectra show two peaks at 21.7° and 24°, which are characteristic of the (110) and (200) lattice planes, respectively, in a crystal region of polyethylene. A peak at 19.5° can be obtained through peak separation analysis using Gaussian function to the XRD spectra, which represents the amorphous region. The crystallinity can be obtained from the following relationship [23]:(1)$$W_{XLPE}=\frac{I_{110}+1.42I_{200}}{I_{200}+1.42I_{200}+0.65I_{a}}$$
where *W_XLPE_* is the crystallinity, *I_a_*, *I*_110_, and *I*_200_ are the diffraction intensity of amorphous, (110), and (200) peaks, respectively.

According to the specific changes in crystal diffraction peaks, there is no obvious change in the position of the diffraction peaks during the whole aging period, indicating that the basic crystal form of each aging stage is the same. The diffraction intensities of the (110) and (200) peaks decrease with aging time after 384 h. In addition, at the end of the middle stage of thermal aging (after 384 h), serrated protrusions appeared in the XRD spectrum; that is, the diffraction intensity increased or decreased near the diffraction peak. With aging time increasing, the number of zigzag distortion protrusions increases and the peak distortion becomes stronger.

#### 3.1.2. Analysis of the Physicochemical Properties

In the FTIR spectra of new cables and cable samples with different on-site aging degrees, it was observed that the absorption peak at wavenumber 2010 cm^−1^ is not sensitive to thermal aging, and its strength is not affected by the degree of thermal oxygen aging, while the carbonyl absorption peak is the opposite. Therefore, it is suitable to calibrate the characteristic 2010 cm^−1^ peak as a reference peak. The carbonyl index is defined as the ratio of carbonyl absorption peak intensity at wavenumber 1720 cm^−1^ to absorption peak intensity at wavenumber 2010 cm^−1^(A1720/A2010) [24].
(2)$$\begin{array}{l} \mathrm{Carbonyl}\;\mathrm{Index}(\mathrm{CI})\\ =\frac{{\mathrm{Absorption}\;\mathrm{at}\;1720\mathrm{cm}}^{-1}(\mathrm{the}\;\mathrm{maximum}\;\mathrm{of}\;\mathrm{carbonyl}\;\mathrm{peak})}{{\mathrm{Absorption}\;\mathrm{at}\;2010\mathrm{cm}}^{-1}} \end{array}$$

The CI change in the cable’s inner insulation during thermal aging is shown in Table 1. The CI can be used to characterize the material thermal aging degree. As shown in Table 1, CI exhibits almost no change from 48 to 96 h (the early aging stage). During 192–384 h (middle aging stage), CI begins to increase due to the consumption of antioxidants and the production of polar products. At the late aging stage (after 384 h), CI increases sharply; this rapid rise in CI can be attributed to the radicals generated by the chain scission of the polymer in the presence of oxygen, meaning that this process is accompanied by chain breaking and the formation of a large number of polar oxidation products. Additionally, the radicals caused by chain breaking could lead to the formation of pairs of alkoxy radicals. The reactions of alkoxy radicals could result in the formation of more polar groups, such as carbonyls, aldehydes, and ketones [24], making the CI increase further.

In order to further analyze the spectral lines, the half-peak width (FWHM) of the crystallization peak and crystallinity during thermal aging are also calculated. The FWHM of the two crystal peaks and crystallinity at each aging stage are also shown in Table 1. According to the Table 1, the FWHM fluctuates firstly at the early aging stage. Then, with the increase in aging time, during 192–384 h, the (110) and (200) peaks start to broaden. The serrated protrusion and broadening of the half-peak width indicates the distortion of the XRD diffraction spectra and the appearance of zigzag protrusion, whereas the broadening of peak shape indicates that dislocation defects have been initiated in the crystal zone, and that some crystal atoms deviate from the original position. At this stage, although the crystallinity does not strongly decrease, the crystal region has been destroyed slightly. When the damage degree of crystal zone is low, the crystallinity does not decrease greatly (48–384 h). Whereas with the aging degree deepening, at the late aging stage, FWHM of (110) and (200) peaks broaden rapidly and the serrated protrusion density increases. With the increase in dislocation defect density in the material, the damage degree of crystal region increases, resulting in the decrease in crystallinity (after 784 h) [25].

### 3.2. High-Voltage FDS Results

The high-voltage FDS of unaged cable sample is shown in Figure 8. In Figure 8, the maximum tanδ of the unaged cable at each detection voltage level can only reach about 1.0 × 10^−3^ (*U*_0_, 0.01 Hz), and tan δ does not increase or decrease regularly with the frequency change. Moreover, the curves show cluster shape and do not separate in the frequency domain.

According to the microscopical physicochemical test results, the aging degrees are divided into three phases: the early, middle, and late aging stages. The high-voltage FDS curves of the cable at the early aging stage (48–92 h) are shown in Figure 9a,b. The FDS curve shapes of the unaged and early aging stage cable are similar, but the tan δ values show an increasing trend. In addition, the tan δ of the early aging stage cable increases as the detection frequency decreases, indicating the slope of the tan δ-f curves increase compared to the unaged cable. However, at this aging time point, the cable is still at the stage of early aging, the increase in dielectric loss is not large, and there is no regular variation in the change in voltage level. Correspondingly, the performance changes are not obvious. As shown in Table 1, the EAB, CI, and crystal structure exhibit almost no change.

The high-voltage FDS curves of the cable at the middle aging stage (192–384 h) are shown in Figure 7 and Figure 10a, respectively. According to Figure 7, the tan δ value continues to increase with the aging degree. In particular, starting from a higher voltage level (*U*_0_), the tan δ-f curves gradually increase as the detection voltage rises at each frequency point, meaning the FDS curves start to show a layered phenomenon. Correspondingly, according to the results in Section 3.1, at this aging time point, the mechanical properties begin to show a declining trend, the polar products increase, and dislocation defects begin to appear in the crystal region.

The high-voltage FDS curves of the cable at the late aging stage (after 384 h) are shown in Figure 11a,b. At the late aging stage, the cable tan δ value continues to increase, up to about a maximum of 180 × 10^−3^ (*U*_0_, 0.01 Hz). The layered phenomenon of the tan δ-f curves becomes increasingly obvious at each detection voltage level. At this stage, on the one hand, the decreased rate of EAB accelerates and rapidly decreases to the attention value (EAB 75%) then the failure value (EAB 50%). On the other hand, chain breaking and oxidation reaction produce more polar groups, and high-density defects in the crystal zone of material lead to a decrease in crystallinity.

According to [15,16], the high-voltage FDS of water-tree-aged cable also increases as the detection voltage rises. In addition, the high-voltage FDS of the water-tree-aged cable also exhibits the hysteretic effect, i.e., when a higher voltage is applied between the sweeps at the same low voltage level, the loss levels from a second measurement sweep (at the same voltage level) are higher than those in the previous sweep. The previous and the second FDS curves of low voltage level (0.25 *U*_0_) before and after a higher voltage (*U*_0_) at the late aging stage (thermally aged 1536 h) are shown in Figure 12. According to Figure 12, the lower voltage detection curves are crossed in frequency domain. This means that, even at the late stage, thermal aging does not cause the hysteresis effect. Therefore, the water treeing and thermal aging can be distinguished by hysteresis phenomenon.

### 3.3. Discussion of Experiment Results

#### 3.3.1. Correlation Analysis of High-Voltage Dielectric and Physicochemical Properties

The XLPE molecular structure is shown in the Figure 13. As seen in Figure 13, XLPE is composed of crystalline region and amorphous region. A main chain can pass through several crystalline and amorphous regions, and the chain is rigidly arranged in parallel with the crystalline region, and curled and irregularly arranged in the amorphous region. The dipolar groups interact with each other via the backbone of the chain, as also indicated schematically in Figure 13. This interaction process may produce the material loss. Here, the reversal of any one dipole must react on relatively distant dipoles through the rigid chain linking them, and it is intuitively plausible to assume that the resulting energy loss constant is independent of the rate of reversal, but dependent on the number of reversed dipoles and their reversal amplitude [26].

From the perspective of polarization, the polarization strength *P* of the material can be considered as a response caused by the electric field strength *E*, that is [27]:(3)$$P=\chi \varepsilon_{0}E$$

The non-aging insulation is non-polar, the dipole density is low, so the dielectric loss of the cable changes little with frequency and detection voltage. In the early stages of aging, antioxidants are not consumed. At this stage, on the one hand, the oxidation reaction will lead to chain breaking (thermal oxidative cracking) of polyethylene molecules. Meanwhile, on the other hand, due to the presence of antioxidants, small molecules will be re-crosslinked, thus, there are few oxidation products and low polarization strength. Therefore, there is no obvious change in the CI. Correspondingly, during this period, although the dielectric loss and curve slope increase slightly, the change is not obvious with the increase in detection voltage and frequency. With the extension of aging time, antioxidants are gradually depleted. At the middle aging stage, the CI starts to increases, indicating that the XLPE molecular chain starts to break and the oxidation reaction begins to occur. From this stage, the increase in polar matter on the one hand leads to the increase in dipole density in the material, which helps to improve the polarization intensity. On the other hand, chain breaking makes dipole reversal easier. Thus, during the middle aging period, from the higher detection voltage, the polarization strength increases, so does the material loss. At the late thermal ageing stage, the CI increases drastically, showing more oxidation polar products. During this period, the dipole density is higher and the dipole rotation is less affected by the rigid main chain.

Furthermore, according to the XRD results, at the beginning of the middle stage of thermal aging, serrated protrusions appear in the XRD spectrum, that is, the diffraction intensity increases or decreases near the diffraction peak. In addition, from Table 1, the half-peak width of XLPE in the middle stage of thermal aging is widened. The above results show that dislocation defects appear in the material at this stage, and some crystal atoms deviate from the original position. At this stage, although the crystallinity does not decrease greatly, the crystal region has been destroyed. Thus, the rigidity of the main chain of XLPE molecule begins to decrease, the probability of dipole reversion increases, and the reversal amplitude also increases. Especially when the external electric field is large, the loss of the cable begins to increase significantly, that is, the cable appears delamination. At the late stage of thermal aging, dislocation defect density gradually increases drastically, and the crystallinity of XLPE begin to decrease significantly, which means the increase in the amorphous region. At this aging time point, the reversion probability of the dipole continues to increase and the inversion angle is larger. There is a stable stratification phenomenon in FDS spectrum and the loss increases.

According to Figure 12, even at the late stage, thermal aging does not cause the hysteresis effect. Some researchers believe that the water tree insulation consists of small water-filled voids separated by crazing zones. When increasing the test voltage, Maxwell mechanical stresses will cause water to penetrate the crazing zones, thus forming electric contact between the elongated water droplets. The electric field at the tip of these conductive channels is enhanced, resulting the dielectric losses increase [28]. However, the opening and closing processes of the channel are gradual [29]. Therefore, in a certain time interval, the water tree channel is not completely closed in the second low voltage detection, resulting in the second tan δ value is greater than the first tan δ value. As previously analyzed, the dielectric property change during thermal aging is mainly caused by polarization and dipole reverse, which is different from the change mechanism of water tree aging, thus there is no hysteresis effect. Therefore, we consider that the hysteresis effect can be used to distinguish the water tree aging from thermal aging.

#### 3.3.2. Discussion on Diagnosis Method

According to the comparison of Figure 8, Figure 9, Figure 10 and Figure 11, that the tan δ-f curves increase with the rising detection voltage for the cable with a certain aging degree. Additionally, the curve presents a layered characteristic in the frequency domain, meaning at any detection frequency point, tan δ (*U*_0_) > tan δ (0.75 *U*_0_) > tan δ (0.5 *U*_0_) > tan δ (0.25 *U*_0_). The ratio of dielectric loss at higher voltage to that at lower voltage must be greater than 1. Therefore, the layered characteristic of the high-voltage FDS curves can be extracted as the quantity to characterize the change in XLPE cable FDS. The layering degree (*L*) is defined as the dielectric loss ratio mean value of each adjacent detection voltage level at each frequency point:(4)$$L_{1}=\frac{1}{n}\sum_{k=1}^{n}\frac{\tan_{0.5U_{0}}\delta \left(f_{k}\right)}{\tan_{0.25U_{0}}\delta \left(f_{k}\right)}$$
(5)$$L_{2}=\frac{1}{n}\sum_{k=1}^{n}\frac{\tan_{0.75U_{0}}\delta \left(f_{k}\right)}{\tan_{0.5U_{0}}\delta \left(f_{k}\right)}$$
(6)$$L_{3}=\frac{1}{n}\sum_{k=1}^{n}\frac{\tan_{U_{0}}\delta \left(f_{k}\right)}{\tan_{0.75U_{0}}\delta \left(f_{k}\right)}$$
where *L*_1_, *L*_2_, and *L*_3_ are the layering degrees of FDS curves between different voltage levels, *n* is the number of detection frequency points, and *f_k_* is the detection frequency.

The *L* matrix of high-voltage dielectric FDS curves of thermal aging cable at various aging stages is shown in Table 2.

According to the layered characteristics comparison and analysis of the thermally aging cable, it can be seen that the layering coefficient *L*_1_, *L*_2_, and *L*_3_ are not all greater than 1 when the cable is not aged and with weaker aging degree. While with the deepening of aging severity, the appearance of polar substances and the change in crystal structure lead to the molecular polarization increase. Therefore, the FDS curve tan δ-f rises with the detection voltage level increase, resulting in the layered phenomenon of FDS curves, so the layered degree *L*_1_, *L*_2_, and *L*_3_ are all greater than 1. Therefore, the layering coefficient matrix of high-voltage FDS curve can be used to diagnose the cable insulation thermal aging.

From the change in the high-voltage FDS with aging time (Figure 8, Figure 9, Figure 10 and Figure 11), the slope (*K*) and integral (*S*) of the curve change distinctly with the aging degree. Moreover, the detection voltage also has a significant effect on these two parameters. The FDS test voltage influence on *K* and *S* at different aging stages are shown in Figure 14. According to Figure 14, *K* and *S* increase with aging time. Especially, when the detection voltage is higher (*U*_0_), the change is more obvious. Therefore, *K* and *S* under *U*_0_ detection voltage can be used as characteristic values to diagnose aging degree.

The changes in *K* and *S* under *U*_0_ detection voltage (*K_U_*_0_*/S_U_*_0_) are also shown in Table 2. In the early aging stage (48–96 h), the *K_U_*_0_ and *S_U_*_0_ values almost do not change; in the middle aging stage (192–384 h), *K_U_*_0_ and *S_U_*_0_ begin to increase; after 384 h, *K_U_*_0_ and *S_U_*_0_ rise sharply. These trends are consistent with the change in CI, FWHM, and crystallinity (in Table 1), EAB (in Figure 5). Therefore, we consider that *K_U_*_0_ and *S_U_*_0_ are suitable characteristics to diagnose the aging degree.

## 4. Conclusions

In this study, the high-voltage FDS (0.25 *U*_0_–*U*_0_, 0.01 Hz–0.1 Hz) of an XLPE cable with different aging degrees and the physical, chemical, and mechanical properties of the insulation materials are tested. In addition, the relationships between these factors are also explored. On this basis, the diagnosis and evaluation method of cable thermal aging based on high-voltage FDS is discussed.

(1) The high-voltage FDS of the cables with a certain degree of thermal aging presents layered characteristics, which is different from low-voltage FDS. The experimental results suggest that the layering degree L of high-voltage FDS is greater than 1, thus the layering degree matrix can be used as the judgment basis for cable aging.

(2) According to the FTIR and XRD results, it is indicated that the increase in polar groups (carbonyls, aldehydes, and ketones, etc.) and crystal regions are produced by molecular bond breakages, causing an increase in the polarization intensity. The above factors drive the layered characteristic; with the increase in aging degree, the layered phenomenon of the aging cables is more obvious. This is due to the sharp increase in polar groups and the decrease in crystallinity, which reduces the rigidity of molecular main chain and makes it easier for dipoles to reverse.

(3) The changes in the tan δ-f curve slope and integral (*K_U_*_0_*/S_U_*_0_) are very sensitive to thermal aging effects, and they also change consistently with the micro-physical characteristics change: CI, FWHM, crystallinity and elongation at break. Therefore, they can be considered as useful diagnostic parameters for estimating the degree of XLPE thermal aging deterioration.

## Figures and Tables

**Figure 1 polymers-14-03519-f001:**
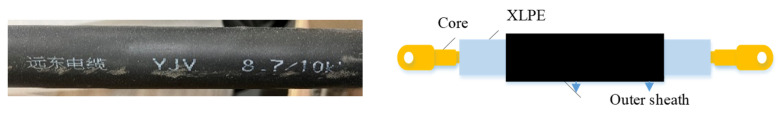
Sample pretreatment diagram.

**Figure 2 polymers-14-03519-f002:**
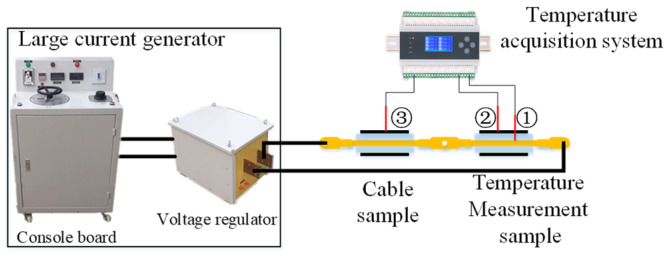
The overall accelerated thermal aging system.

**Figure 3 polymers-14-03519-f003:**
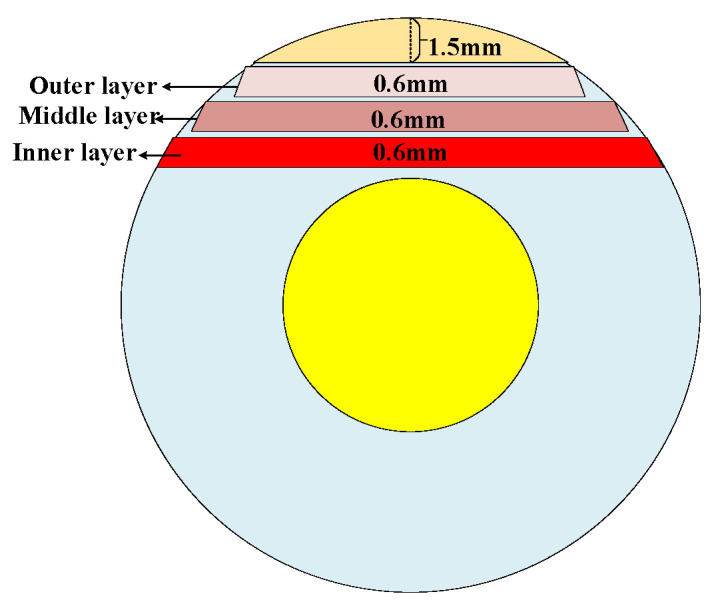
Schematic diagram of cable crosscutting process.

**Figure 4 polymers-14-03519-f004:**
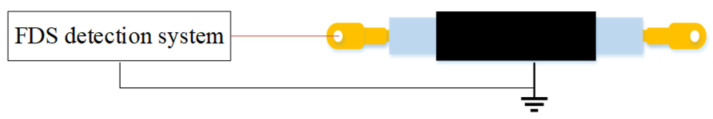
FDS measurement platform and connection diagram.

**Figure 5 polymers-14-03519-f005:**
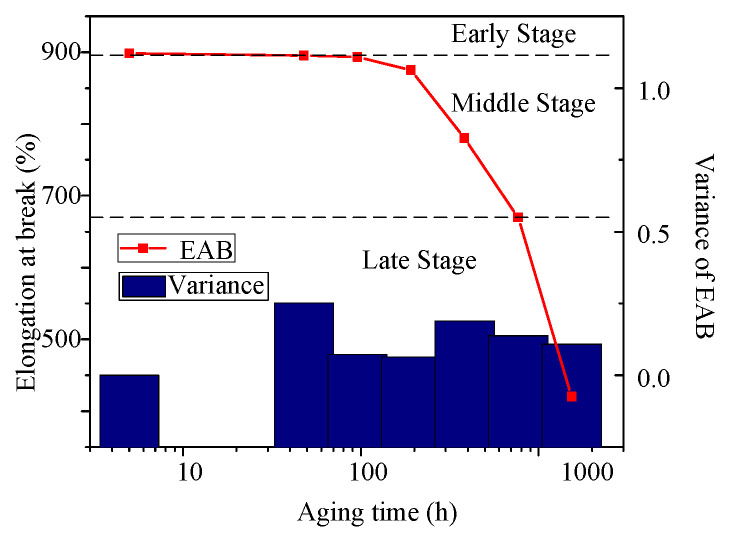
EABs and variance versus aging time.

**Figure 6 polymers-14-03519-f006:**
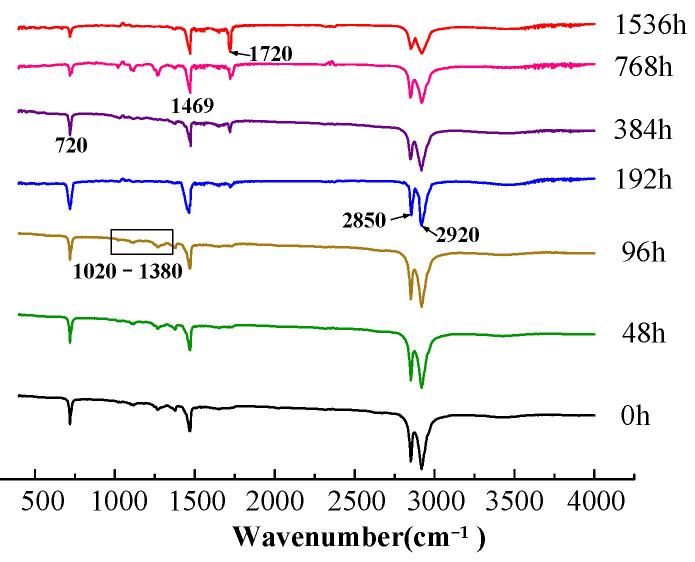
FTIR spectra of XLPE versus thermal aging time.

**Figure 7 polymers-14-03519-f007:**
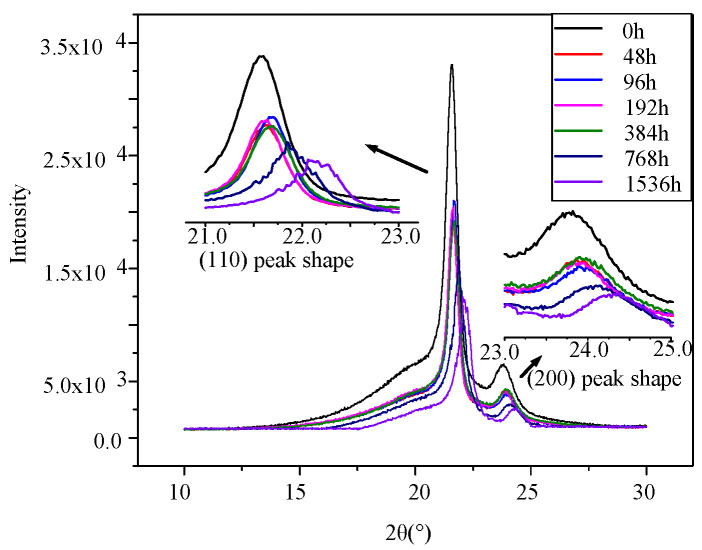
XRD curves versus aging time.

**Figure 8 polymers-14-03519-f008:**
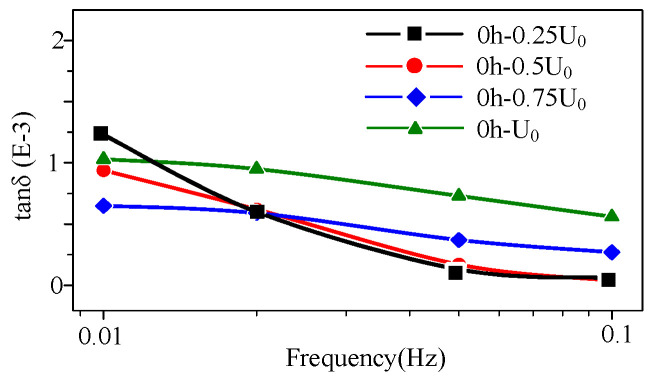
High-voltage FDS of the unaged cable (aging time 0 h).

**Figure 9 polymers-14-03519-f009:**
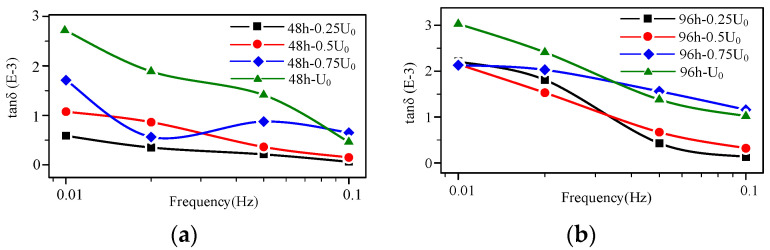
High-voltage FDS of cable at the early aging stage: (**a**) aging time 48 h; (**b**) aging time 96 h.

**Figure 10 polymers-14-03519-f010:**
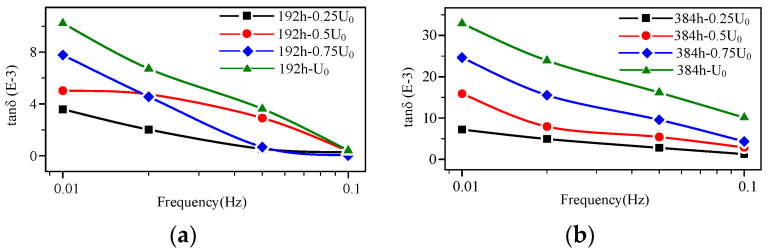
High-voltage FDS of cable at the middle aging stage: (**a**) aging time 192 h; (**b**) aging time 384 h.

**Figure 11 polymers-14-03519-f011:**
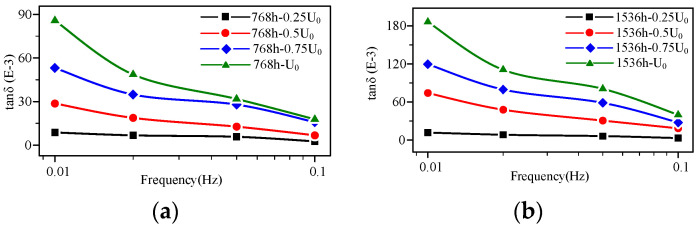
High-voltage FDS of cable at the late aging stage: (**a**) aging time 768 h; (**b**) aging time 1536 h.

**Figure 12 polymers-14-03519-f012:**
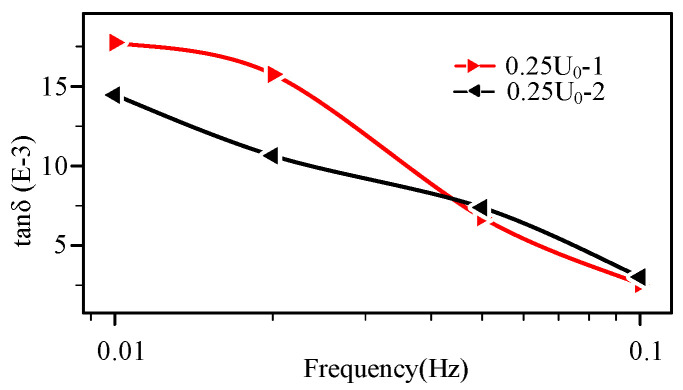
The previous and second lower voltage measurement before and after a higher voltage (at the late aging stage, aging time 1536 h).

**Figure 13 polymers-14-03519-f013:**
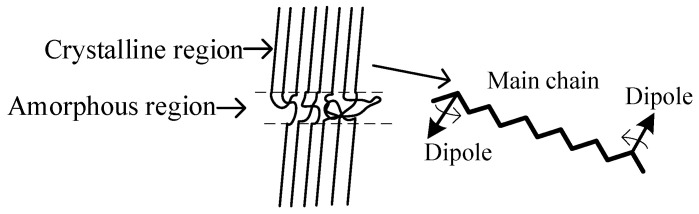
The arrangement of XLPE molecular structure and its main chain.

**Figure 14 polymers-14-03519-f014:**
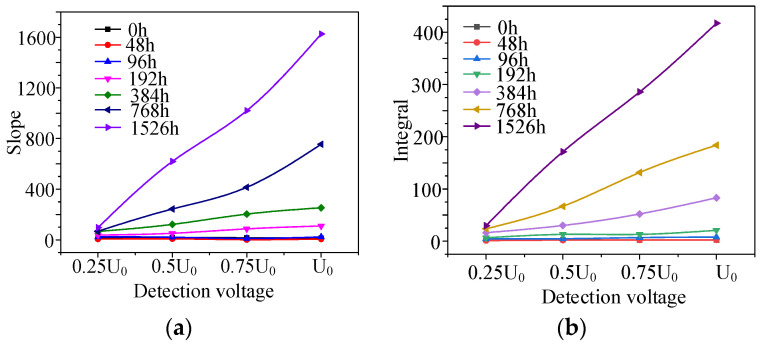
Slope *K* and integral *S* of different aging time versus detection voltage (1536 h): (**a**) slope *K*; (**b**) integral *S*.

**Table 1 polymers-14-03519-t001:** CI, XDR data of each aging stage.

Aging Time	CI	FWHM		Crystallinity	Aging Stage
		(110)	(200)		
0 h	1	0.5372	1.4970	45.69%	–
48 h	1.05	0.5795	1.5668	45.95%	Early Stage
96 h	1.03	0.5436	1.5752	46.03%	
192 h	1.33	0.5630	1.5859	45.70%	Middle Stage
384 h	1.79	0.6188	1.7012	45.03%	
768 h	2.64	0.6921	1.7708	44.78%	Late Stage
1536 h	4.31	0.7334	1.9287	44.56%	

**Table 2 polymers-14-03519-t002:** Layering coefficients, *K_U_*_0_ and *S_U_*_0_, versus aging time.

Aging Time	Aging Phenomenon			Aging Degree		Aging Stage
	*L* _1_	*L* _2_	*L* _3_	*K_U_* _0_	*S_U_* _0_	
0 h	0.92	0.81	1.29	12.56	2.20	–
48 h	2.65	0.93	0.98	6.23	2.48	Early Stage
96 h	0.62	2.42	0.98	22.33	7.84	
192 h	1.08	0.40	5.95	109.11	20.98	Middle Stage
384 h	1.93	1.74	1.85	253.12	83.11	
768 h	2.53	2.12	1.22	755.33	183.84	Late Stage
1536 h	5.4	1.69	1.39	1625.89	417.34	

## Data Availability

Not applicable.

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
