# Peer review of "High-Voltage FDS of Thermally Aged XLPE Cable and Its Correlation with Physicochemical Properties"

_polymers, 2022, doi:10.3390/polym14173519_

Round 1

Reviewer 1 Report

The paper presents results of a study of  high voltage Frequency Domain Spectroscopy of thermally aged XLPE cable and its correlation with physicochemical properties. The paper presents the investigating the influence of the thermal aging on the XLPE cable, and the relationships between the macroscopically high voltage dielectric and the microscopically physicochemical properties are also elucidated. Obtained results show that after a certain aging time, the high voltage FDS curves of cable have the layered characteristics, and this phenomenon is more obvious as the aging degree increases. Besides, the slope and the integral of the high voltage FDS curves rise with the aging time. According to the experimental results, an on-site diagnostics method of cable insulation thermal aging based on high voltage FDS was studied.

Dear authors. Thank you very much for interesting paper about application of FDS in case of high voltage cable, which is common used to estimate a water content in paper insulation of high voltage transformers. I have some comments and questions, which should be included and answered in the paper.

Questions and suggestions:

1. Introduction chapter presents fundamental information about XLPE, what plays main role as high voltage insulation for power cables. Next FDS method is described. I propose to extend some information about FDS as tool which is used in case of power transformers.

2. Experimental Setup chapter presents used samples of XLPE. Please complete information what kind of cable was used – new or worked unit if possible.

3. Fig. 9 – I can see that tan(delta) decreases with aging time increase. How is it possible? Does temperature help to reduce tan(delta)? Please explain because I did not find the explanation if possible.

4. Fig. 9, 10, 11 – to better read, I suggest to use the same range of tan(delta) on the figures if possible.

Generally, it is very interesting paper, which open new application of known FDS method to high voltage cable. I think, the paper will be ready to be published after minor correction.

Reviewer 2 Report

The paper is devoted for thermally aged cable investigations. The topic is generally interesting, however the paper contain unexplained places (below) and need major revisions.

Why FDS investigations were performed only in frequency range 0.01-0.1 Hz? Why aging was performed

at 135 C?

Line 252:’’ tandelta not changes with frequency’’, however in Fig. 8 this parameter is dependent on frequency.

Line 423: ’’polar groups’’, what is the nature of such polar groups?

DSC investigations of your samples can be useful.

Conclusions should be rewritten in more informative way.

All abbreviations should be explained by first using, for example’’ XLPE’’.

All typos should be corrected, for example line 175 ’’cm-1’’. Measurements units and numbers should be written separately, for example line 177 ’’192h’’.

English need minor revisions.

Round 2

Reviewer 2 Report

Authors make proper corrections according to reviewer remarks and I

suggest publish the paper as it is.